# Perineal Hernia Mesh Repair Using Only the Perineal Approach: How We Do It

**DOI:** 10.3390/jpm13101456

**Published:** 2023-09-30

**Authors:** Emil Moiș, Florin Graur, Levente Horvath, Luminița Furcea, Florin Zaharie, Dan Vălean, Septimiu Moldovan, Nadim Al Hajjar

**Affiliations:** 13rd Department of Surgery, University of Medicine and Pharmacy “Iuliu Hațieganu”, Croitorilor Street, No. 19–21, 400162 Cluj-Napoca, Romania; drmoisemil@gmail.com (E.M.); hunorlevente.horvath@gmail.com (L.H.); luminita.furcea@yahoo.com (L.F.); florinzaharie@yahoo.com (F.Z.); valean.d92@gmail.com (D.V.); septimiu1995@yahoo.com (S.M.); na_hajjar@yahoo.com (N.A.H.); 2Regional Institute of Gastroenterology and Hepatology “O. Fodor”, Croitorilor Street, No. 19–21, 400162 Cluj-Napoca, Romania

**Keywords:** perineal hernia, rectal surgery, mesh repair

## Abstract

Perineal hernia is a rare complication of rectal surgery. Different types of surgical approach have been described, but none of them have proven their superiority. Although there are many methods of closing the defect, we selected two cases to present from a series of five cases, in which the perineal hernia was successfully resolved surgically using only the perineal approach. The reconstruction of the perineal floor and closure of the defect were performed using a synthetic polypropylene mesh. The significance of this Technical Note article lies in the fact that we describe, step by step, a surgical technique for perineal hernia using just a perineal approach.

## 1. Introduction

Perineal hernia (PH) is defined as the protrusion of the intraabdominal viscera through a defect in the pelvic floor into the perineal space [1]. Acquired PH is a rare but documented complication of extensive pelvic surgery, particularly abdominoperineal resection (APR) for rectal cancer, and pelvic exenteration. The reported incidence is as low as <1% in symptomatic cases and approximately 7% in asymptomatic cases. However, in recent years, an increase in the incidence has been observed, probably because of the increased use of neoadjuvant chemoradiation [2,3].

PH generally occurs within the first two years after surgery, with a median interval of 10 months. It is most often accompanied by a bulging mass or pain associated with discomfort that worsens when standing or sitting. Skin erosion, urinary dysfunction, and bowel obstruction are among the complications that can accompany PH [3,4].

The treatment is surgical, but because of the lack of evidence and the existence of only a small number of studies on this topic due to the low incidence of PH, there is no consensus on optimal surgical management.

Various surgical approaches have been described for repair, including perineal, abdominal, or a combination of the two, and abdominal procedures can be performed using either an open or laparoscopic approach [1,5].

In the next section, we describe a surgical technique used to resolve two cases of perineal hernia by using only a perineal approach.

## 2. Materials and Methods

Over the course of the last three years, 5 patients were admitted with perineal hernia. In all cases, the surgical technique used is the one presented in this article. The 2 cases presented include intraoperative pictures and computed tomography (CT) scans performed within our service. In all cases, mesh repair, using the perineal approach, was taken into consideration. The study was approved by the Ethics Committee of the Regional Institute of Gastroenterology and Hepatology “O. Fodor”, Cluj-Napoca, Romania. All the patients have signed written consent.

## 3. Results

### 3.1. Case Presentation

#### 3.1.1. Case 1

The first case was that of a 75-year-old woman with symptomatic PH after an abdominoperineal resection for rectal cancer. In 2019, the patient was diagnosed with T4bN0M0 inferior rectal cancer and underwent neoadjuvant chemoradiotherapy. Surgery consisted of an APR with a terminal sigmoidostomy, a total hysterectomy with a bilateral salpingo-oophorectomy, and a colpectomy. Eighteen months postoperatively, the patient presented with worsening bulge symptoms associated with discomfort, particularly while sitting. On physical examination, the patient had no fever and was hemodynamically stable. On inspection of the perineum, a large bulging mass, covered by skin, was observed, and the patient described pain during palpation (Figure 1).

As the patient was symptomatic, surgery was indicated. A transperineal hernia repair was performed with the patient in the lithotomy position. After a vertical incision was made over the defect, the hernial sac was identified, containing viable small bowel content (Figure 2).

The small bowel contents were reduced after opening the hernial sac (Figure 3).

A dissection of the hernial sac was carefully performed, after which the excess was excised, and the sac was closed. A polypropylene mesh was then inserted across the defect (above the levator ani muscles) and anchored to the endopelvic fascia and adjacent muscular structures with 1.0 non-absorbable polypropylene sutures (Figure 4).

Hemostasis was achieved, a subcutaneous surgical drain was placed, and the subcutaneous and skin tissues were closed. Postoperative recovery was uneventful, and the patient was discharged on postoperative day 5. No recurrence was observed during the last follow-up visit (one and a half years).

#### 3.1.2. Case 2

The second case was a 57-year-old woman who presented to our clinic complaining of a perineal bulge and discomfort while sitting or walking. In 2020, she was diagnosed with inferior rectal adenocarcinoma (T4bN0M0) and underwent neoadjuvant chemoradiotherapy (nCRT), followed by surgery consisting of a laparoscopic APR with a terminal sigmoidostomy. Upon inspection of the perineum, a large perineal bulge was noted at the level of the postoperative scar (Figure 5).

The hernia measured 10 × 15 cm and was covered with skin. It had an elastic consistency and was reducible, with no pain described during palpation. Further workup included laboratory tests and a thoracoabdominal and pelvic CT scan, which confirmed the presence of an extensive perineal hernia containing small bowel content. A perineal approach was chosen for repair, a vertical incision was made over the hernia, and the postoperative scar tissue was excised. Once the hernial sac was incised and opened, the small bowel content was reduced (Figure 6 and Figure 7).

After the dissection and excision of the excess hernial sac, the parietal peritoneum was mobilized up to the levator ani muscles and was then closed with simple interrupted sutures. The defect was then covered and closed with a tailored synthetic polypropylene mesh and fixed to the ischial tuberosity and sacral bone (on the levator ani muscles) using helical titanium tacks (ProTack Covidien, Minneapolis, USA) and 2.0 non-absorbable polypropylene sutures (Figure 8).

With the pelvic floor defect closed, the subcutaneous and skin tissues were closed, and a subcutaneous surgical drain was inserted. Figure 9 shows a schematic representation of how the mesh was placed.

The immediate postoperative period was uneventful, and the patient was discharged on postoperative day 5. The one-and-a-half-year follow-up showed no signs of relapse (Figure 10) during the clinical examination or on a CT scan. (Figure 11).

#### 3.1.3. Surgical Technique

Both surgeries were performed under general anesthesia, with the patient in the lithotomy position to maximize the field of view. The incision is made over the defect, taking some precautionary measures for the underlying tissues, especially for hernias in which the sac is thinner, in order not to injure the small bowel. Removal of the postoperative scar tissue is not mandatory, but we strongly recommend this because of the positive cosmetic effect. After the incision of the hernial sac, its contents must be adequately explored for their viability, and then reduced. Mobilization of the parietal peritoneum should be performed, especially in larger defects, to adequately close the hernial defect in order to avoid contact between the mesh and the small intestine. The excess sac should be removed, as it increases the risk of seromas. However, in cases of larger defects, it can be used as a reinforcement layer. Afterwards, the placement of the synthetic mesh should always be fixed below the levator ani muscles, as this allows for better closure of the defect, limiting the excess space for a possible recurrence. In order to further minimize the risk of recurrence, non-resorbable sutures should be placed. If the adjacent structures have a higher density (such as the pelvic floor), tacks should be preferred over sutures, as they offer better control. After the closure of the pelvic floor, subcutaneous sutures, as well as skin sutures, are applied. Subcutaneous drainage is mandatory, especially when the excess sac is not removed, to further minimize the risk of seromas.

Postoperative recovery was uneventful, and the patient was discharged on postoperative day 6. No recurrence was observed during the last follow-up visit (one and a half years—Figure 10)

## 4. Discussion

Perineal hernia is defined as a pelvic floor defect through which the intraperitoneal or extraperitoneal contents may protrude. It is a rare complication, with an incidence of less than 1% of symptomatic PH. However, the true incidence is unknown; it could be higher because of non-reported asymptomatic cases [1,3]. Risk factors that play a role in the development of PH include radiotherapy, smoking, diabetes, obesity, malnutrition, and collagen disorders [5,6,7].

In recent years, the incidence of PH has shown an upward trend, which could be secondary to increased utilization of nCRT, decreased adhesions due to increased use of laparoscopy, increased survival of patients with rectal cancer, or more aggressive surgical resections [4,8]. In response to the increase in PH, emphasis has been placed on investigating different reconstruction techniques of the perineum after surgery to reduce perineal wound healing complications [2]. Different methods have been described for reconstruction, including primary/direct closure, mesh reconstruction, gluteal and rectus abdominis flaps, or combinations of these techniques. The selection of the repair method is pivotal in preventing the recurrence of the perineal hernia, especially in cases of nCRT. Primary closure has been documented to exhibit a recurrence rate of 50%, whereas the utilization of a mesh reduces this rate to 20% [9,10,11].

The BIOPEX study, the only multicenter randomized control trial focusing on perineal reconstruction using biological mesh after an extra-levator approach (eAPR), showed that PH occurs significantly less often after mesh reconstruction than after primary closure. The BIOPEX study used an acellular non-crosslinked mesh derived from porcine dermis, sutured posteriorly either to the sacrum or the coccis or laterally to the remnant of the levator muscle; anteriorly, it was fixed to the transverse perineal muscles. The utilization of biological mesh for pelvic floor closures following an eAPR for rectal cancer in patients who had received preoperative radiotherapy showed no significant difference when compared to the group that underwent primary perineal wound closure at 30 days. Moreover, there were no notable variations in perineal wound healing or quality of life observed at other postoperative time intervals between the two randomization groups for up to 1 year after the surgery. Notably, the rate of freedom from perineal hernia at 1 year was significantly higher in the biological mesh group, reaching 87%, as opposed to the primary closure group, which had a rate of 73% [4]. When comparing the type of mesh used, comparable recurrence rates were observed, whether it be biological or synthetic. Bertrand et al. observed a 47.1% recurrence rate for biological mesh and 40% for synthetic. Given its higher cost, the use of biological mesh should be reserved for high-risk populations, which may include individuals with a prolonged history of chronic pelvic sepsis, highly irradiated tissues, enterocutaneous fistulas, or bowel injuries sustained during dissection [3,12].

Regarding flap reconstructions, the use of pedicled muscle flaps, such as vertical rectus abdominis musculocutaneous (VRAM) flaps, offers successful healing, and seems to be the best flap reconstruction solution. The pedicled VRAM flap has found utility in various applications for challenging periabdominal wounds and is associated with fewer perineal complications compared to primary wound closures. Furthermore, it exhibits lower perineal morbidity and boasts excellent long-term survival outcomes, with success noted up to 10 years postoperatively. Additionally, it surpasses gracilis flap reconstruction in terms of complication rates. However, the applicability of the VRAM flap can be restricted by various factors such as patient positioning, previous abdominal surgeries, scarring, and the number of required ostomies. V-to-Y advancement flaps, in contrast, mitigate additional abdominal wall morbidity when compared to the VRAM flap, and provide a less bulky option for flap coverage. While local tissues can promote effective healing and deter perineal hernias, they are disadvantaged by their proximity to the radiated surgical field, potentially leading to increased wound healing complications. Another effective alternative described is the pedicled gracilis muscle flap, which offers a wide range of rotation without limiting postoperative mobility or ambulation. However, it may be inadequate for larger pelvic defects due to its size [13].

Most commonly, symptomatic PH patients present with a bulging mass associated with pain or discomfort, which is worse in standing or sitting positions. The diagnosis is clinical, and CT or MRI can be performed during the oncological follow-up; however, the positioning of the patient may reduce hernia detection rates. The main complications of PH include skin erosion, urinary dysfunction, and bowel obstruction [14]. Due to its low incidence and a lack of studies on PH, the optimal treatment remains a controversial topic. Because of the lack of evidence, the surgical approach is mainly decided by the surgeon’s preference and experience. A perineal, abdominal, combined abdominoperineal, or even laparoscopic approach can be used to treat PH. A series of case reports have stated that the perineal approach should be the first choice for uncomplicated PH. Regarding patient positioning in the perineal approach, opinions vary throughout the literature; some authors described successfully using the prone position as well [5].

The abdominal approach provides better exposure and the ability to perform another procedure at the time of repair and perform a diagnostic exploration of the abdomen to assess oncological recurrence. Moreover, if patient selection is adequate, it seems that the abdominal approach has a smaller recurrence rate than the perineal approach. It is preferred if the reduction of the hernial sac content is difficult, given that dissection and control may be easier. When this approach is used, the patient should be positioned in the Lloyd-Davies position, and abdominal exploration is mandatory regardless of whether the approach is open or minimally invasive. All herniated organs, typically the small bowel and/or bladder, are carefully dissected out of the hernia sac. The hernia sac itself is excised to establish the anatomical boundaries of the defect, although it may be left in place if it is small or dissection is deemed risky. The urogenital organs, when present, are situated anteriorly, while the bony structures of the pelvic outlet, often with varying degrees of overlying pelvic floor musculature, constitute the posterolateral and remaining aspects of the hernial defect. Throughout this procedure, great care is taken to identify and protect the ureters, bladder, and prostate (or vagina) when applicable. If there is any available omentum left, it should be positioned over the mesh, and if feasible, the pelvic peritoneum should be closed to prevent the inclusion of the bowel. If mesh is used to cover the defect towards the posterior and lateral aspects, the sacral promontory and endopelvic fascia provide suitable locations for suture placement. Anteriorly, the pubic rami serve as anchor sites [2,3,14].

The abdominal approach can also be pursued laparoscopically, and in recent years, it has become increasingly popular, slowly replacing the open approach. When laparoscopy is preferred, the patient is positioned in the lithotomy position, and usually, four access ports are needed in order to achieve optimal work conditions. The mesh can be fixed in this case either by using surgical tacks or sutures, but a combination of the two can also be used. The mesh has to be fixed to the promontory or sacrum and to the pelvic side walls with a minimum of 1 cm overlap [15,16]. No study has yet proved the superiority of either approach; however, laparoscopy has several advantages. First, it allows for better dissection of the hernial sac; it also provides good access for mesh positioning, and an omentoplasty can also be performed. It is also likely that the laparoscopic approach will provide similar advantages as observed in laparoscopic versus open bowel resection, particularly regarding reduced length of hospital stay and less postoperative pain. The primary drawbacks associated with a laparoscopic approach continue to include the risk of extensive adhesiolysis, especially following pelvic surgery, along with an elevated risk of mesh infection when a colostomy or ileostomy is present in the operative field. Although it has some disadvantages as well, laparoscopy has been used in many cases and has been proven to be a safe and feasible approach to perineal hernia repair and will probably be the gold standard in the future [17]. Ultimately, the most suitable approach will depend on the unique circumstances of each patient and the specific features of their perineal hernia. Given the limited available data due to the low incidence and the complexity of perineal hernias, it is too early to definitively conclude that one approach is the unequivocal treatment of preference.

Similarly, various techniques have been described to repair pelvic floor defects, including omentoplasty, synthetic or biological mesh repair, musculocutaneous rotation flaps, and free fascia lata flaps. Omentoplasty can be an excellent method due to its straightforward nature, but it might not always be accessible, adequate, or sturdy enough to effectively cover the defect. When a perineal hernia occurs following a sacrectomy, the posterior and caudal defect tends to be substantial, and the pelvic floor might be too lax for alternatives other than mesh reconstruction or tissue restitution securely anchored to the pelvic bone. Synthetic mesh repair is an excellent choice, but it raises concerns about the risk of fistula formation or the development of chronic mesh infections that might necessitate removal. It is also not advisable when the surgical site is contaminated. In cases involving complex or complicated hernias, such as those associated with abscesses or tumors, or when contamination occurs during surgery, myocutaneous flap reconstruction may be considered a more suitable option. Whether the flap consists of muscular tissue alone or requires myocutaneous transfer depends on the condition of the skin covering the hernia [6]. The reconstruction of the perineum with myocutaneous flaps frequently necessitates the involvement of a plastic surgeon due to their complexity, leading to extended operating times. In some situations where mesh insertion is not feasible due to inadvertent fecal contamination, bladder mobilization to create a cystopexy has been described as a method to bridge the defect. Recently developed bioprosthetic materials, while more expensive, offer potential advantages including resistance to bacterial infection, autologous tissue remodeling and revascularization, reduced risk of intestinal adhesion, and integration into irradiated tissue. Closing perineal wounds primarily following an eAPR for a rectal cancer procedure is linked to a significant risk of wound infection and dehiscence. When the levator muscles are excised, only fatty tissue and skin remain, necessitating closure under tension, which subsequently results in a high occurrence of wound-related issues.

The best data on mesh choice shows that synthetic mesh is still likely to be the best option, and it is our choice for treating these cases as well; however, due to the small number of cases, further research is necessary to determine the best choice [2,6,18].

## 5. Conclusions

We reported two cases from a series of five with PH after APR for rectal cancer that were resolved using only a perineal approach, which in terms of recovery is better for the patients. The reconstruction of the perineal floor and closure of the defect were performed using a synthetic polypropylene mesh. The technique described can be used successfully.

## Figures and Tables

**Figure 1 jpm-13-01456-f001:**
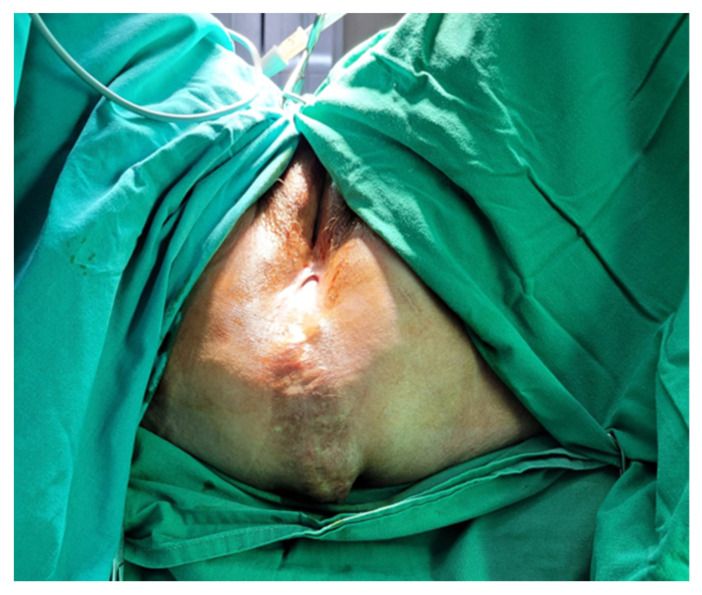
Clinical examination of the perineum.

**Figure 2 jpm-13-01456-f002:**
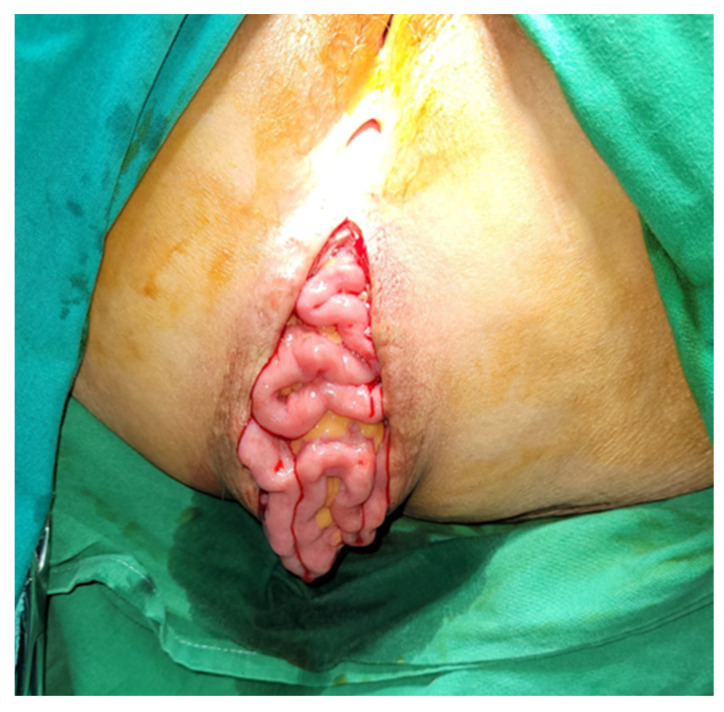
The hernial sac with the abdominal contents.

**Figure 3 jpm-13-01456-f003:**
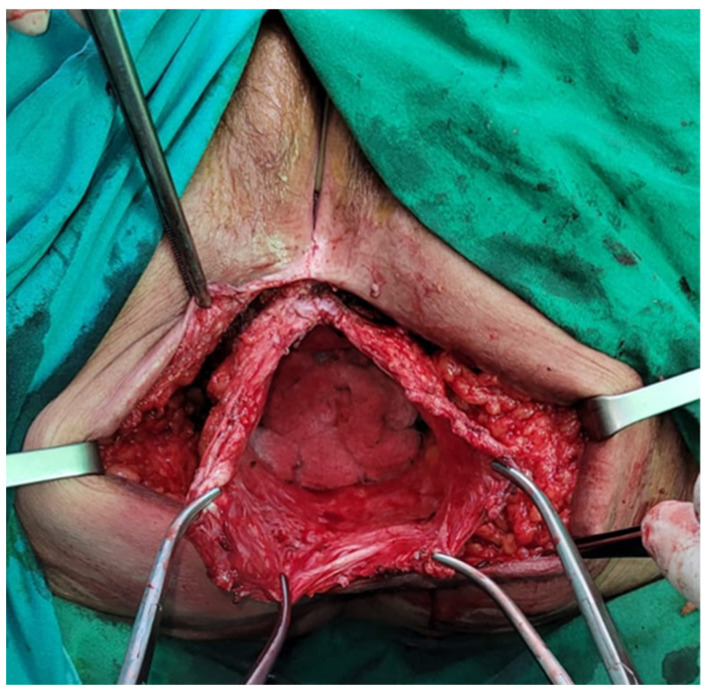
Reduced hernial sac contents.

**Figure 4 jpm-13-01456-f004:**
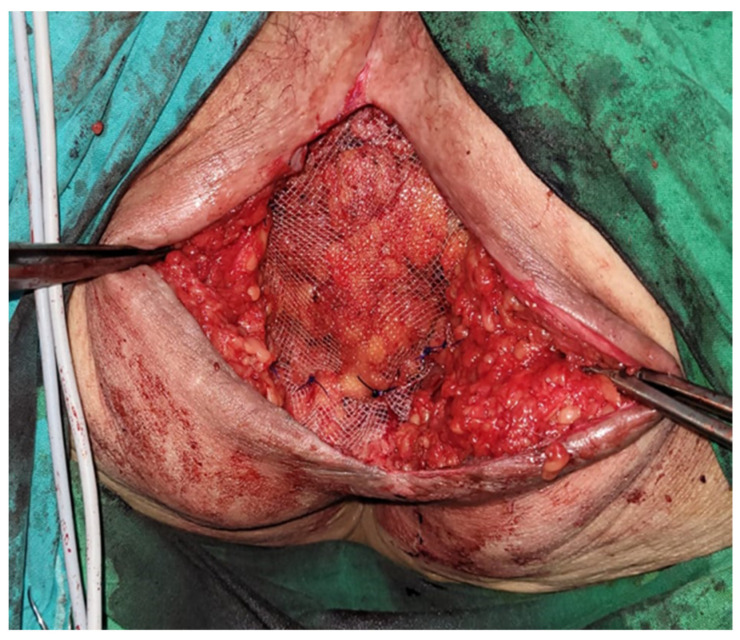
Placement of the polypropylene mesh.

**Figure 5 jpm-13-01456-f005:**
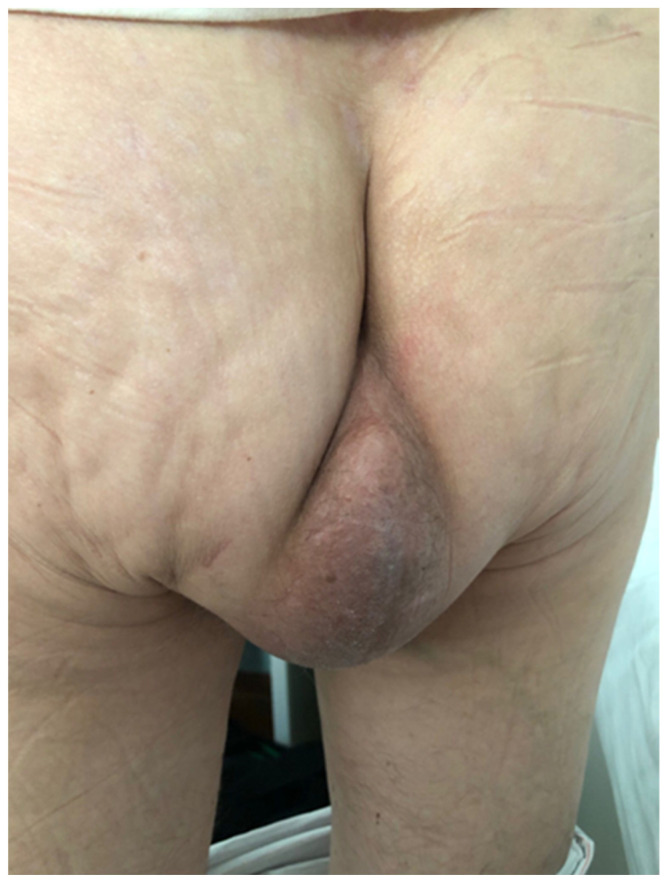
Inspection of the perineum.

**Figure 6 jpm-13-01456-f006:**
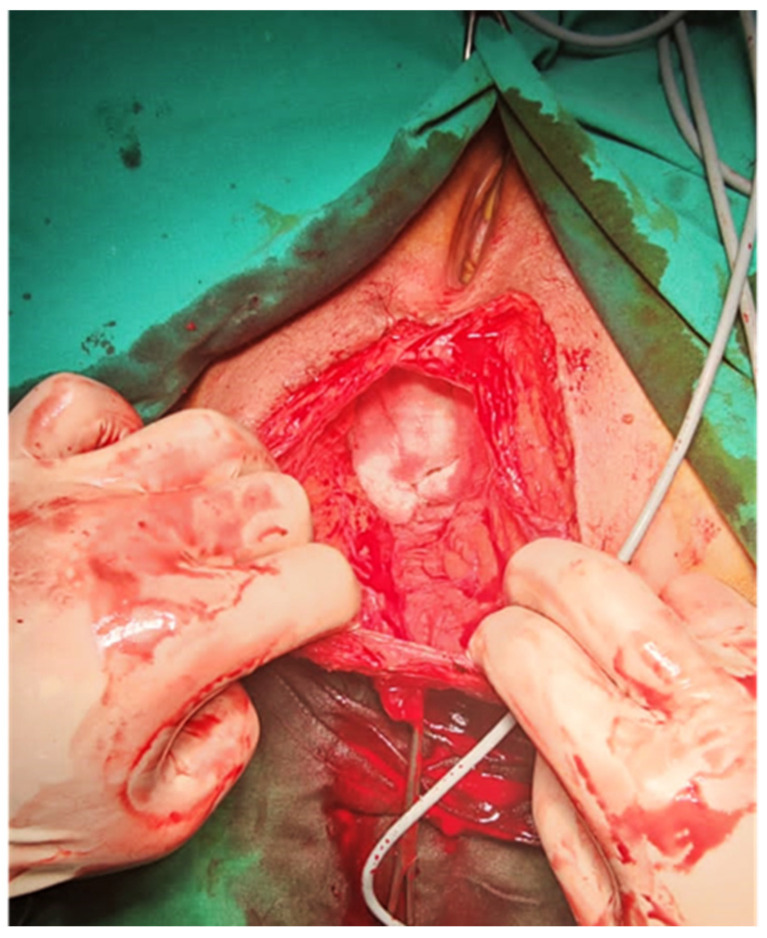
Incision and dissection of the hernial sac.

**Figure 7 jpm-13-01456-f007:**
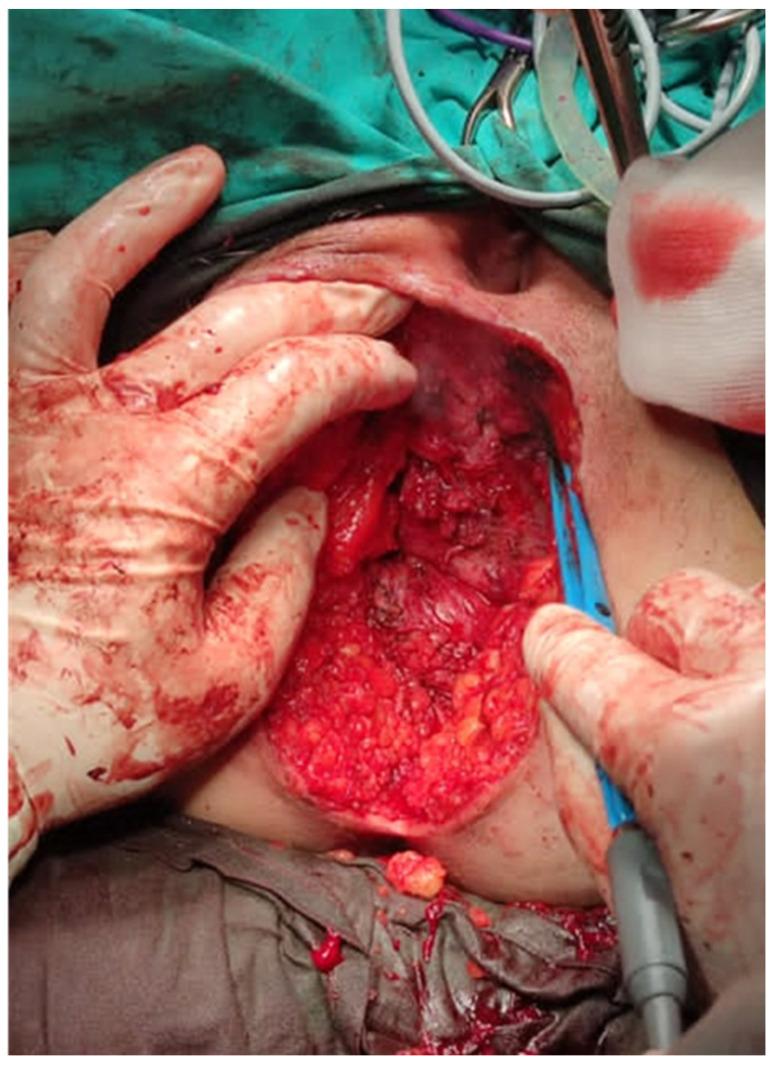
Reduction of the hernial contents.

**Figure 8 jpm-13-01456-f008:**
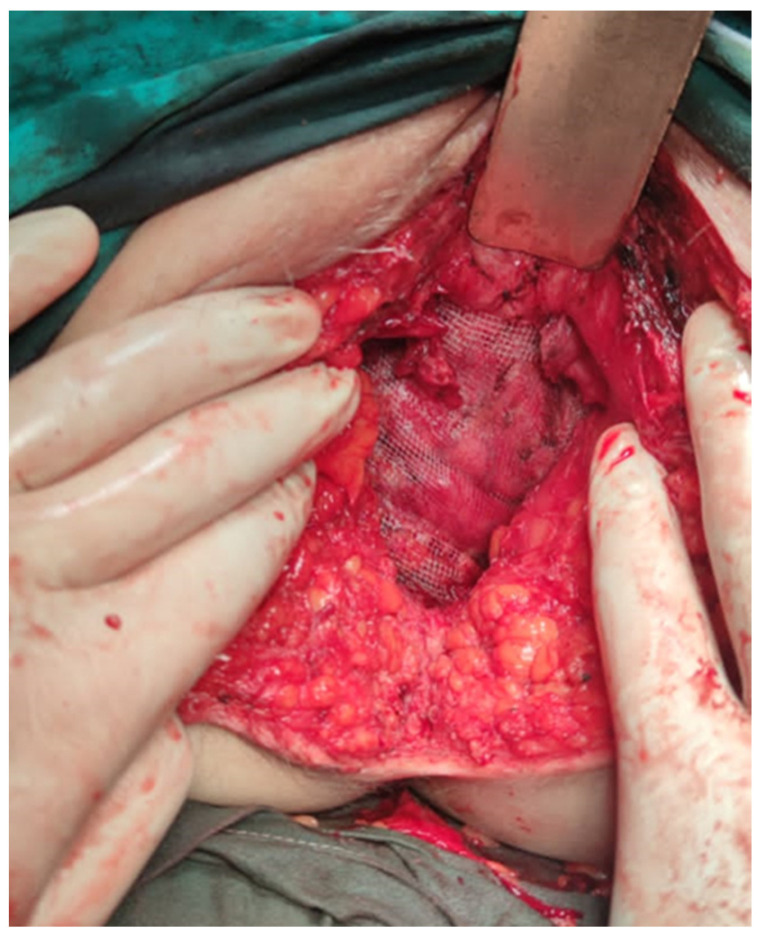
Placement of the tailored polypropylene mesh.

**Figure 9 jpm-13-01456-f009:**
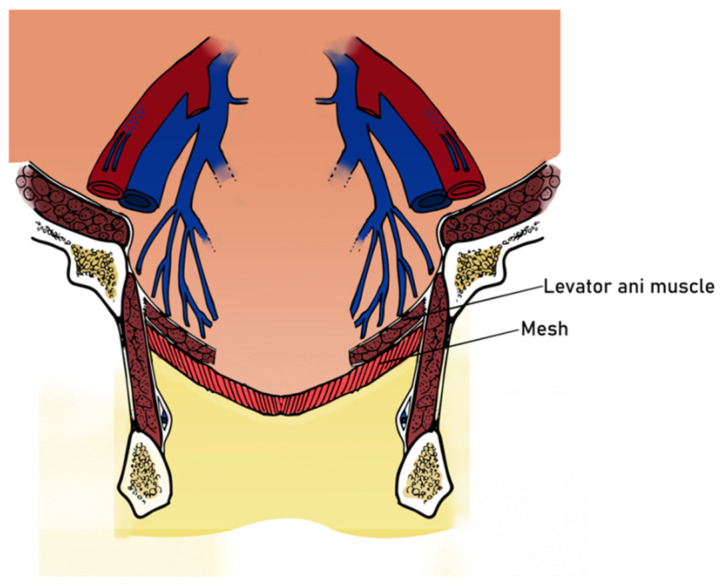
A schematic representation of how the mesh was placed.

**Figure 10 jpm-13-01456-f010:**
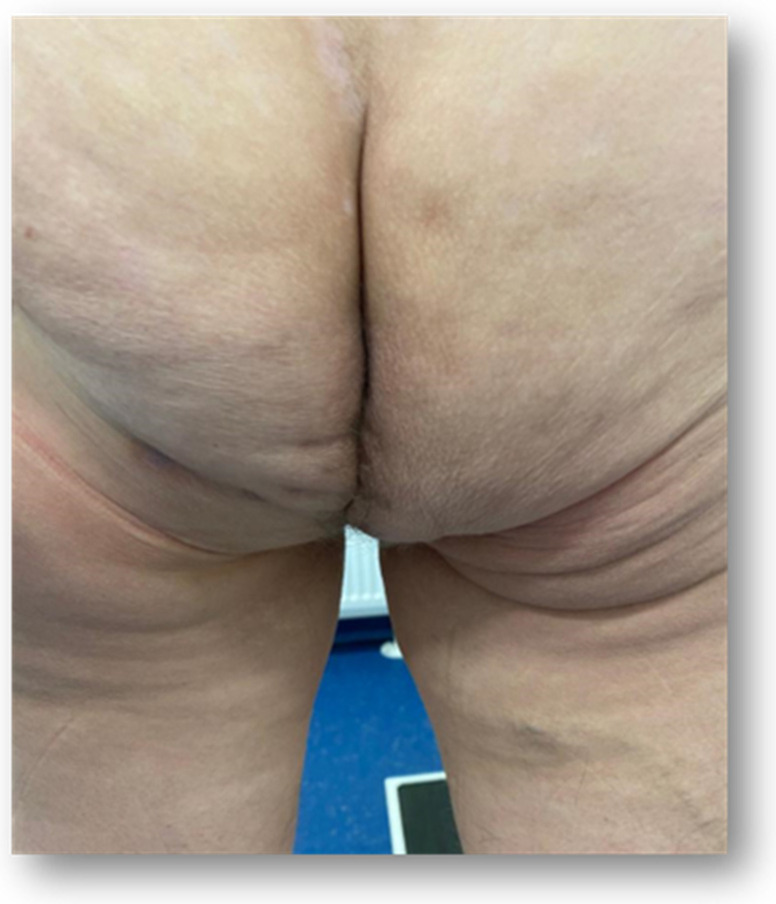
One-and-a-half-year follow-up—the clinical aspect.

**Figure 11 jpm-13-01456-f011:**
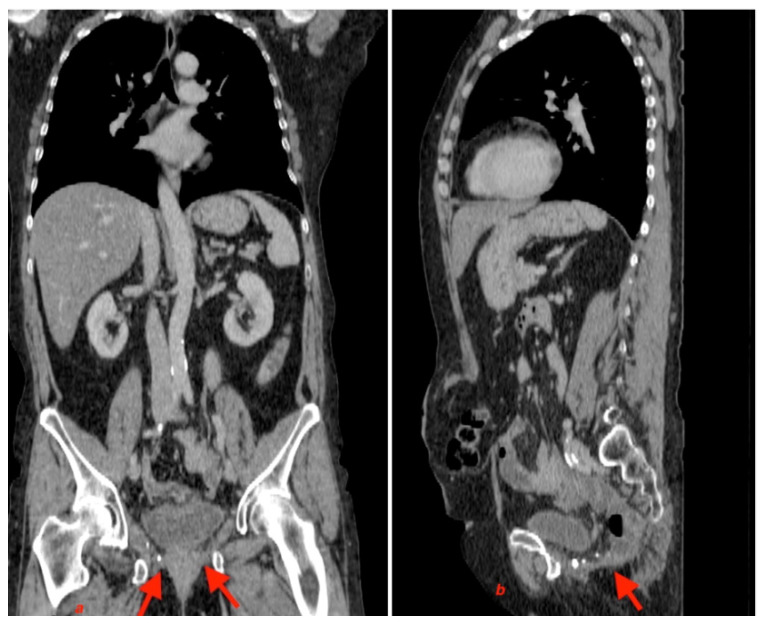
(**a**) CT scan coronal section; (**b**) CT scan sagittal section (arrows showing the place of the mesh).

## Data Availability

Not applicable.

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
