# Peer review of "Perineal Hernia Mesh Repair Using Only the Perineal Approach: How We Do It"

_jpm, 2023, doi:10.3390/jpm13101456_

Round 1
Reviewer 1 Report
Case 1 : I think it's useful to know the duration of the follow-up.
Surgical technique : Did you record any postop complications? Did you record, in relation to polypropylene mesh, enteric fistulas?
Reviewer 2 Report
A small series of patients with a history of abdominoperineal resection (APR) and subsequent development of perineal hernia is the subject of the authors' retrospective study. They present a description of a surgical solution of the problem.
Here in details my observations:
- Line 78-79: please specify how long was the follow up after surgery.
- Line 72, 106-107 and figure n#9: the authors define the level of placing of the synthetic mesh as “above the levator ani muscles”, in my opinion it should be better defined as “on the levator ani muscles”, Clinically speaking, the term 'above' reminds of a more cranial plane than the anatomical structure mentioned. Instead, in this case, the mesh is anatomically placed under the muscles. In this regard, I would like to know if the authors have ever considered the opportunity to place the mesh beyond the levator ani muscles, inside the abdominal cavity, instead of placing it below. For example, in repairing ventral abdominal hernia the retro-muscular position of the mesh is assumed worldwide as the first choice due to its correlation with a lower rate of recurrences. Any analogy with the topic of this article? The risk of bowel adhesion to the mesh and the choice of absorbable implant for this use are currently being debated.
- Line 128-129: “Removal of the postoperative scar tissue is not mandatory”. I am definitely in contrast with this affirmation. Whenever feasible, the pre-existing scar tissue should be excised in order to improve wound healing and decrease the local complication rate.
- The authors stated that they treated 5 patients with the technique, but only two of them have been followed up. Why? What about the remaining three patients?
- The study presents some limitations: a very limited study population, no data about age, BMI, performance status, and comorbidities of the patients, no stratification, and no clear endpoints. As an article whose aim is supposed to be investigating a single procedure, it can be allowed. Nonetheless, in my opinion, a comparison should be made between that procedure and the many others reported in the literature, with a mention of at least the pros and cons of each one. To focus on the proper clinical scenario, the readers need more than just citing the alternatives. Expanding the Discussion chapter and examining detailed aspects and adverse effects of the procedure is something I strongly recommend.
The article is well-written and offers interesting insights into a procedure for a well-known complication that can occur after abdominopelvic resection. However, the presentation of the procedure without a proper discussion on pro and cons and about alternatives fails to provide the reader a balanced report, that should be the primary goal of such type of publications.
I suggest that the authors include patient data in Methods and add a comparison to other surgical alternatives in the Discussion chapter.
